# Occult Vancomycin-Resistant *Enterococcus faecium* ST117 Displaying a Highly Mutated *vanB*_2_ Operon

**DOI:** 10.3390/antibiotics12030476

**Published:** 2023-02-27

**Authors:** Antonella Santona, Elisa Taviani, Maura Fiamma, Massimo Deligios, Hoa M. Hoang, Silvana Sanna, Salvatore Rubino, Bianca Paglietti

**Affiliations:** 1Department of Biomedical Science, University of Sassari, Viale San Pietro 43/b, 07100 Sassari, Italy; 2Dipartimento di Scienze della Terra, Dell’ambiente e della Vita—DISTAV—Università degli Studi di Genova, Corso Europa 26, 16132 Genova, Italy; 3Clinical-Chemical Analysis and Microbiology Laboratory, San Francesco Hospital, 08100 Nuoro, Italy; 4Department of Microbiology & Carlo Urbani Center, Hue University of Medicine & Pharmacy, Hue 530000, Vietnam; 5Division of Microbiology and Virology, University Hospital of Sassari, 07100 Sassari, Italy

**Keywords:** occult *vanB*_2_-VREfm, Vitek2 system, WGS, cgMLST

## Abstract

Rare information is available on clinical *Enterococcus faecium* encountered in Sardinia, Italy. This study investigated the antimicrobial susceptibility profiles and genotypic characteristics of *E. faecium* isolated at the University Hospital of Sassari, Italy, using the Vitek2 system and PCR, MLST, or WGS. Vitek2 revealed two VanB-type vancomycin-resistant *Enterococcus faecium* (VREfm) isolates (MICs mg/L = 8 and ≥32) but failed to detect vancomycin resistance in one isolate (MIC mg/L ≤ 1) despite positive genotypic confirmation of *vanB* gene, which proved to be vancomycin resistant by additional phenotypic methods (MICs mg/L = 8). This *vanB* isolate was able to increase its vancomycin MIC after exposure to vancomycin, unlike the “classic” occult *vanB*-carrying *E. faecium*, becoming detectable by Vitek 2 (MICs mg/L ≥ 32). All three *E. faecium* had highly mutated *vanB*_2_ operons, as part of a chromosomally integrated Tn*1549* transposon, with common missense mutations in VanH and VanB_2_ resistance proteins and specific missense mutations in the VanW accessory protein. There were additional missense mutations in VanS, VanH, and VanB proteins in the *vanB*_2_-carrying VREfm isolates compared to Vitek2. The molecular typing revealed a polyclonal hospital-associated *E. faecium* population from Clade A1, and that *vanB*_2_-VREfm, and nearly half of vancomycin-susceptible *E. faecium* (VSEfm) analyzed, belonged to ST117. Based on core genome-MLST, ST117 strains had different clonal types (CT), excluding nosocomial transmission of specific CT. Detecting *vanB*_2_-carrying VREfm isolates by Vitek2 may be problematic, and alternative methods are needed to prevent therapeutic failure and spread.

## 1. Introduction

Nosocomial *Enterococcus faecium* (*E. faecium*) infections are caused by a specific hospital-associated (HA) *E. faecium* subpopulation, capable of prolonged survival outside the human body, contributing to cross-contamination through direct contact between patients and healthcare workers. This leads to local outbreaks, as well as increased length of hospitalization, mortality rate, and healthcare costs [1,2]. HA-*E. faecium* infections are characterized by ampicillin resistance (AREfm) and often by multidrug resistance (MDR) [1]. 

In their pathway towards MDR, HA-AREfm can acquire vancomycin resistance determinants under antibiotic pressure, becoming the worrisome vancomycin-resistant *E. faecium* (VREfm), which is responsible for more severe infections in high-risk patients, especially from intensive care unit (ICU) and surgery wards, with limited therapeutic options [3].

The most common vancomycin resistance determinant in clinical settings is *vanA*, but in the last few years, *vanB* clones have been emerging, overtaking *vanA* in some European countries [4,5]. Both *vanA* and *vanB* determinants are associated with mobile genetic elements (MGEs)—the *vanA* gene cluster as a part of Tn3 family transposon Tn*1546*, and *vanB* as part of Tn*1547* or Tn*1549*/Tn*5382*-like conjugative transposons, which play a crucial role in disseminating vancomycin resistance intra- and interspecies [6,7].

The VanA- and VanB-type glycopeptide resistance expressions are regulated by a two-component regulatory system, VanR_A_S_A_ and VanR_B_S_B_, respectively, composed of a membrane-associated sensor kinase (VanS_A_ and VanS_B_) and a cytoplasmic response regulator (VanR_A_ and VanR_B_) [8]. The cytoplasmic response regulator VanR acts as a transcriptional activator controlling the transcription of *vanR_A_S_A_* regulatory and *vanHAXYZ* resistance genes by PR and PH promoters’ activation in the *vanA* operon, and *vanR_B_S_B_* and *vanY_B_WH_B_BX_B_* genes by activating the PR_B_ and PY_B_ promoters in the *vanB* operon [8]. Unlike *vanA*, which confers inducible resistance to high levels of both glycopeptides vancomycin and teicoplanin, the *vanB* operon is not induced by teicoplanin, and it encodes for a variable level of inducible vancomycin resistance, in some cases below the European Committee on Antimicrobial Susceptibility Testing (EUCAST) breakpoint of 4 mg/L. Thus, *vanB*-containing isolates can go undetected by phenotypic methods, as recently described for the Vitek2 instrument [9]. 

Further, *E. faecium* isolates can harbor silenced *van* operons resulting in vancomycin susceptible clones, thus escaping the routine antimicrobial tests [10]. These types of isolates, named Vancomycin-Variable *E. faecium* (VVEfm), showed various vancomycin resistance silencing mechanisms, including deletions of the *vanRS* regulatory genes [11], deletions or disruptions of resistance genes [12,13,14], and mutations in resistance genes [15]. During vancomycin therapy, VVEfm can revert to vancomycin-resistant phenotypes through various mechanisms, causing treatment failures, silent transmission, and outbreaks [10,11,12,13,14,15]. 

Genomic surveillance through whole-genome sequencing (WGS) has proved its usefulness for accurate and effective infection control practices, for detecting variants of VREfm that are missed by conventional screening techniques, and for molecular epidemiology studies also tracing intra-hospitals dissemination of nosocomial clones [16].

In this study, we aimed to characterize, through phenotypic and molecular methods, *E. faecium* isolated from 2013–2018 in the University Hospital of Sassari, Italy. WGS was used to characterize the *vanB* determinants of the first described *vanB*-VREfm isolates in our hospital.

## 2. Results

### 2.1. Phenotypic and Genotypic (PCR) Characterization of E. faecium Isolates 

Overall, *39 E. faecium* isolates collected from patients from ICU (23%), Surgery (54%), and Medicine (23%) wards of the University Hospital of Sassari were analyzed. A description of the characteristics of the isolates is given in Table 1. The antibiotic resistance pattern was similar among all isolates regardless of isolation date, source, and ward. The majority (92%) were ampicillin resistant (AMP MIC ≥ 32 mg/L), as well as resistant to high-level streptomycin (92%), high-level gentamicin (80%), imipenem (92%), and ciprofloxacin and erythromycin (100%). All isolates were susceptible to both linezolid and tigecycline. Two isolates (SSM5811 and SSM5812) showed resistance to vancomycin by the Vitek2 system exhibiting VanB phenotypes with VAN MIC = 8 mg/L and ≥32 mg/L, respectively, and TEC MIC ≤ 0.5 mg/L (Table 1). The two isolates were collected from the same patient (from a wound swab and a central vascular catheter 24hr later) who was admitted at the ICU in May 2018.

The presence of the *vanB* gene in the two VREfm isolates SSM5811 and SSM5812 was confirmed by PCR. A third isolate (SSM5777), isolated in 2017 from ICU, also carried the *vanB* gene, even displaying susceptibility to glycopeptides (VAN MIC ≤ 1 mg/L and TEC MIC ≤ 0.5 mg/L) by Vitek2 using the gram-positive susceptibility test cards P592 and P658.

The three *vanB E. faecium* isolates were then tested by MicroScan using gram-positive card AB33 (Table 2) and by disk diffusion; both methods were also able to detect the vancomycin resistance in the occult SSM5777 isolate (Table 2). As a result of disk diffusion for vancomycin susceptibility in the SSM5777 isolate, a fuzzy inhibition zone edge was observed in the SSM5777 isolate. Despite the zone diameter of 12 mm, colonies were found within the inhibition zone, so the isolate was reported as resistant according to the notes of the EUCAST clinical breakpoint. 

A further investigation of the three *vanB*-VREfm isolates was conducted by placing the isolates on BHI broth with increasing concentrations of vancomycin (4, 10, 25, 50, 100 mg/L). All three isolates were able to grow at all tested concentrations at 37 °C after 24 h, including at 100 mg/L after 48 h of incubation. Following exposure to 100 mg/L vancomycin, glycopeptide susceptibility was retested, growing the isolates in the absence of vancomycin. This enabled the Vitek2 system to detect the vancomycin resistance in SSM5777 as well. Glycopeptides MICs were determined by both the Vitek2 system and MicroScan before exposure to vancomycin, and by Vitek2 following exposure to vancomycin, as shown in Table 2. 

### 2.2. Multi Locus Sequence Typing (MLST)

Genetic analysis by MLST, which characterizes *E. faecium* using the sequences of internal fragments of seven housekeeping genes (*atp, ddl, ghd, purK, gyd, pstS,* and *adk*), revealed seven different sequence types (STs) among the isolates, most belonging to ST117 (n = 18) and ST78 (n = 12), followed by ST916 (n = 3), ST80 (n = 2), ST780 (n = 2), ST54 (n = 1), and ST74 (n = 1) (Table 1). Most isolates, including the *vanB* positive isolates, belonged to ST117 (46%) found in all wards, mainly in the ICU, whereas the ST78 *E. faecium* predominated in the surgery ward. The ST117 (9,1,1,1,1,1,1) was first detected in 2013, while the ST80 (9,1,1,1,12,1,1), which differs from ST117 at a single locus (*gyd*) by eBURSTv3 analysis, appeared in 2017 (Table 1).

### 2.3. WGS In Silico Analysis

Ten genomes of *E. faecium* isolates were selected for WGS according to hospital associated STs. Five ST117—including the *vanB*-VREfm, two ST916, and one isolate from ST80, ST78, and ST780—were sequenced by an Illumina platform and assembled to the scaffold level. The accession numbers, type of data, and status of the sequences are listed in Appendix A. The presence of the *vanB*_2_ gene was confirmed in VREfm isolates, as part of a chromosomally integrated Tn*1549* transposon. The Tn*1549* assembled sequence resulted in identical SSM5811 and SSM5812 isolates sharing 99.97% of sequence identity, with the Tn*1549* sequence of *E. faecium* isolate E7654 from Utrecht, the Netherlands (Accession no. LR135324.1), and *E. faecium* AUS0004 (Accession no. CP003351.1). Compared to the *vanB*_2_ gene cluster of AUS0004 (VAN MIC ≥ 16 mg/L and TEC MIC = 2 mg/L) used as a reference strain, our VREfm displayed single nucleotide mutations leading to amino acid substitutions in VanS (D105N), VanW (S204F), VanH (G55S/A231G), and VanB (S23A_I151M_M308L_M321V) proteins (Figure 1). The assembled Tn*1549* of the SSM5777 isolate showed 99.98% sequence identity with those of isolates SSM5811 and SSM5812, while the *vanB*_2_ operon showed a slight genetic variation (99.89%), having a wild type *vanS* gene and different point mutations leading to amino acid substitutions in VanW (T61I_S94P) (Figure 1). 

The SSM5777 was re-sequenced and analyzed after exposure to 100 mg/L of vancomycin, showing the same operon mutations observed before vancomycin exposure.

Moreover, among the ten *sequenced E. faecium,* other resistance genes were detected in silico by ResFinder. The most common were genes conferring acquired resistance to aminoglycosides [*ant(6)-Ia* (n = 8), *aph(3′)-III* (n = 9), *aac(6′)Ii (9),* and *aac(6’)-aph(2′′)* (n = 5)], macrolide-lincosamide-streptogramin B [*msrC* (n = 10) and *ermB* (n = 8)], trimethoprim [*dfr*F (n = 2) and *dfr*G (n = 1)], phenicols [*cat* (n = 1)], and tetracycline [*tet*(M) (n = 2) and *tet*(L) (n = 1)] (Appendix A), while fluoroquinolones resistance was associated with mutations in the quinolone resistance-determining region of topoisomerase genes leading to amino acid substitution in *gyrA* (Ser83-Tyr) and *parC* (Ser80-Ile). For these ten isolates, the phenotypic antibiotic susceptibility patterns aligned with the resistant genes identified (Appendix A).

### 2.4. CgMLST Phylogenetic Analysis 

Core genome MLST analysis revealed that the ST117 isolates belonged to four different complex types (CTs). VSEfm isolates were from CT130 and CT132, and occult *vanB*_2_-VREfm (SSM5777) was from CT798. A new CT was assigned to the *vanB*_2_-VREfm isolates detected in the same patient; they were clonal (single nucleotide polymorphism, SNP = 7) and clustered in Cluster 1 (Figure 2). A putative nosocomial transmission of VSEfm ST916 CT131 was found in the surgery ward in 2013 and clustered in Cluster 2 (Figure 2). The ST80 (9,1,1,1,12,1,1), erroneously designed as a single locus variant of ST117 (9,1,1,1,1,1,1) by eBURSTv3, was genetically more related to isolates from ST78 (15,1,1,1,1,1,1), and hence it is a double locus variant.

## 3. Discussion

Hospital settings provide a reservoir for MDR pathogens; thus, understanding molecular resistance mechanisms and epidemiology of circulating strains is crucial for preventing infection, including the emergence and dissemination of VREfm, which is of particular clinical concern [17]. 

In this study, we performed phenotypic and genotypic characterization of *E. faecium* isolates recovered during 2013–2018 from patients recovering in the main wards of the University Hospital of Sassari. Our findings evidenced three isolates characterized by harboring highly mutated *vanB*_2_ operons that were associated with variable vancomycin MICs representing, to our knowledge, the first description of *vanB*-VREfm in our hospital and in Sardinia. 

Notably, one *vanB*_2_-VREfm isolate was undetected by Vitek2 (occult VREfm), as previously described by other authors [9], while it proved phenotypically vancomycin resistant by alternative methods. Interestingly, unlike the “classic” occult *vanB*-VREfm [9], this isolate was able to increase its vancomycin MIC after exposure to vancomycin, as previously described for *vanA*-VVEfm [12,13,14] and stealthy *vanB*_2_*-E. faecium* isolates [15].

These types of *E. faecium* going undetected by routine phenotypic testing are at risk of silent spread and are currently considered a developing threat because they can revert to a vancomycin-resistant phenotype upon exposure to vancomycin, causing treatment failures. This ability is typical of *vanA E. faecium* [12,14,18]; however, Hashimoto et al. described that Japan HA-*E. faecium* isolates harbor stealthy *vanB*_2_ operons producing mutated VanW and VanB_2_ proteins, which regained a resistant phenotype on exposure to vancomycin thanks to a mutation in the VanS sensor [15]. Our occult *vanB*_2_ isolate showed three mutations in the VanB_2_ protein that were the same as in the other two *vanB*_2_VREfm. Two of these mutations, V321M and M308L, were described to be involved in the decrease of vancomycin MIC in *vanB*_2_-VREfm isolates from Japan and Taiwan, respectively [15,19]. The presence of a VanS mutation (D105N) located in the same periplasmic domain [20] in reverted Japanese isolates [15] may be responsible for the higher vancomycin MIC in our VREfm compared to the occult VREfm, lacking this mutation.

Moreover, the analysis of the *vanB*_2_ operon highlighted additional mutations in VanW and VanH proteins, including a A231G missense mutation in VanH, which occurs in an amino acidic residue essential for its dehydrogenase activity [21]. Considering this, we hypothesized that our *vanB*_2_-VREfm may represent reverted-stealthy *vanB*_2_ *E. faecium,* and that vancomycin exposure may lead to similar mutations in the occult *vanB*_2_-VREfm, as previously demonstrated in Japanese isolates [15]. However, re-sequencing of the *vanB*_2_ operon after vancomycin exposure did not detect additional mutations, suggesting an alternative mechanism for the increased vancomycin MIC. Further studies are needed to confirm this hypothesis and to clarify the role of single mutations in the modulation of the *vanB*_2_ operon. 

The Tn*1549* transposons detected in different years showed high nucleotide identity with common and univocal mutations, suggesting a common ancestral source.

Moreover, most *E. faecium* isolates, including the VSEfm, showed a consistently multidrug-resistant phenotype and genotype over the years. In sequenced isolates, an analysis of acquired antimicrobial resistance genes revealed that aminoglycosides and macrolide resistances were associated with the carriage of the *aph(3′)*-IIIa and/or *aac(6′)*-Ii, and/or *aac(6′)*-Ie-*aph(2′′)*-Ia genes and the *msr(C)* and/or *ermB* genes, respectively. 

Furthermore, despite the low number of isolates analyzed in this study, the polyclonal nature of nosocomial *E. faecium* STs is clearly shown. 

Our results highlighted that *vanB*_2_-VREfm and nearly half of VSEfm analyzed were from the hospital-associated (HA) ST117/CC17 or CladeA1 [22]. These STs are frequently associated with nosocomial outbreaks in Europe [23,24,25] and have also been reported in Italian hospitals [26], along with the other HA-STs that were detected here (ST78, ST916, ST80, and ST780).

Finally, cgMLST subdivided ST117 into different CTs excluding the nosocomial transmission of a specific CT. However, the presence of ST117 over the years and its association to vancomycin resistance determinants compared to the other STs also capable of escaping phenotypic detection makes ST117 even more dangerous, confirming its high potential for diffusion. 

## 4. Materials and Methods

### 4.1. E. faecium Isolates, Antimicrobial Susceptibility 

Thirty-nine *E. faecium* strains isolated from patients admitted at the University Hospital of Sassari, Italy during 2013–2018, all part of the strains collection of the Microbiology Unit of the Biomedical Sciences Department of the University of Sassari, were included in the study. The isolation was performed at the Division of Microbiology and Virology of the University Hospital of Sassari, Italy using Slanetz Bartley or Blood Agar media (Microbiol UTA, Cagliari, Italy). Species identification was determined by Maldi-TOF Mass Spectrometry (Bruker, Billerica, MA, USA). After identification, antimicrobial susceptibility testing was conducted by Vitek2 using the AST cards P592 and P658 (bioMerieux, Marcy-l’Étoile, France), which are specifically for gram-positive *Staphylococcus* spp., *Enterococcus* spp., and *Streptococcus agalactiae*. For *Enterococcus* spp., among the antibiotics tested, we report results for ampicillin (AMP), high-level gentamicin (GEN), high-level streptomycin (STR), ciprofloxacin (CIP), imipenem (IMI), erythromycin (ERY), linezolid (LNZ), teicoplanin (TEC), vancomycin (VAN), and tigecycline (TGC) following the European Committee on Antimicrobial Susceptibility Testing (EUCAST) guidelines [27].

Antimicrobial susceptibility testing was also performed by MicroScan autoSCAN-4 using the AB33 Cards (Beckman Coulter Inc., CA, USA), following the EUCAST guidelines [27]. Glycopeptide susceptibility was additionally tested by the disk diffusion method [27] with *E. faecalis* ATCC 29212 as the control.

### 4.2. Molecular Characterization

#### 4.2.1. Screening for *vanA*, -*B*, -*C* Genes and MLST

Bacterial DNA was extracted with the DNeasy Blood & Tissue Kit (QIAGEN, Inc., Valencia, CA, USA). The presence of *vanA*, *vanB*, *vanC1*, and *vanC2/C3* genes was found by Multiplex-PCR as previously described [28]. MLST was carried out on all isolates with the standard primers included in the *E. faecium* MLST scheme [29]. In addition, purified amplicons with DNA clean and concentrator^TM^-5 columns (Zymo Research, Irvine, CA, USA) were sequenced at the LMU Sequencing Service (Munich, Germany). Obtained sequences were analyzed using Geneious Pro 4.8.4 (http://www.geneious.com/ accessed on 1 March 2017). Allelic profiles and STs were assigned according to the *E. faecium* MLST database (http://pubmlst.org/efaecium/ accessed on 16 December 2018) and included in CCs based on eBURSTv3 analysis (http://eburst.mlst.net/ accessed on 16 December 2018) used to assign isolates to genetic complexes and to evaluate the genetic relationships of the STs.

#### 4.2.2. WGS and In Silico Analysis

WGS was performed on ten selected *E. faecium* isolated from the surgery (SSM5539, SSM5540, SSM5541, SSM5544, SSM5551, SSM5552, and SSM5775) and ICU (SSM5777, SSM5811, and SSM5812) ward. Chromosomal DNAs, extracted using the DNeasy Blood & Tissue Kit (QIAGEN, Inc., Valencia, CA, USA), were quantified by the NanoDrop Microvolume Spectrophotometer (ThermoFisher, MA, USA) and submitted to WGS using a HiScanSQIllumina platform (Porto Conte Ricerche Srl, Tramariglio, Italy). Preparation of the DNA libraries was performed with the Nextera XTDNA Sample Preparation Kit (Illumina Inc., San Diego, CA, USA) and sequenced using the HiScanSQ (Illumina Inc., San Diego, CA, USA) with 93bp × 2 paired-end reads.

Generated sequences were assembled de novo into contigs using the software Velvet version 1.2.10 [30]. Contigs were reordered against the reference genome of *E. faecium* AUS0004 (Accession no. CP003351.1) using Mauve. Whole-genome alignment was done using the Artemis Comparative Tool (ACT) and MUMmer. The NUCmer tool of the MUMmer software was used to search individual gene sequences within the genomes. All ten genomes were submitted to the RAST platform for annotation (https://rast.nmpdr.org/ accessed on on 15 May 2021. The assembled genomes were subjected to the online search tools MLST 2.0 (Achtman MLST scheme) and ResFinder 4.0 available at the Center for Genomic Epidemiology (CGE) (http://www.genomicepidemiology.org/ accessed on 30 June 2021. The *vanB*_2_ operon was resequenced using specific primers designed on the AUS0004 reference strain (CP003351) and shown in Appendix A. Amplicons were purified, sequenced, and analyzed as described above.

### 4.3. Phylogenetic Analysis by cgMLST

The Ridom SeqSphere+ version 6.0.2 [31] whole-genome typing platform was used to infer phylogenetic relatedness of the isolates based on 2554 loci of the *E. faecium* cgMLST v1.1 and *E. faecium* MLSTv1.0 schemes, with cgMLST complex type/cluster-alert distance set as 20.

## 5. Conclusions

In this work, we highlighted a highly mutated *vanB*_2_-VREf ST117 occult isolate by Vitek2 with the ability to increase its vancomycin MIC after exposure to vancomycin. This isolate was found to have similarities to both *vanA*-VVEfm and stealthy *vanB*_2_-*E. faecium.*

The risk of non-detection of *vanB*_2_-VREfm by the Vitek2 system is of great concern, and *E. faecium* should always be processed using an alternative method to Vitek2 to avoid therapeutic failure and *vanB*_2_-VREfm spread.

Stricter preventive measures, such as isolation of infected/colonized patients and more accurate disinfection procedures, can reduce the risk of dissemination of these coverts and all HA-VREfm/VSEfm clones.

## Figures and Tables

**Figure 1 antibiotics-12-00476-f001:**
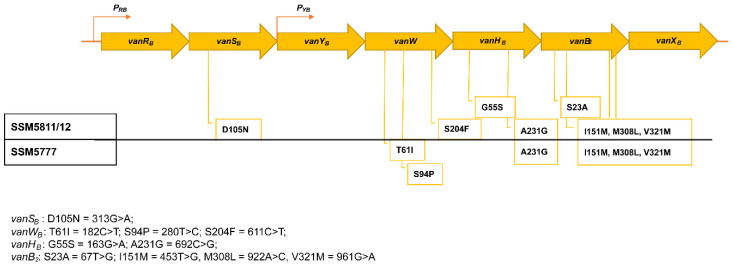
Schematic representation of *vanB*_2_ operon in VREfm and in occult isolate, with the amino acid substitutions framed and in line with the corresponding regulation and resistance genes. For each *van* gene, the amino acid replacements (D = Asp, N = Asn, T = Thr, I = Ile, S = Ser, P = Pro, F = Phe, G = Gly, A = Ala, M = Met, L = Leu, and V = Val) corresponding to nucleotide mutations are shown below on the left.

**Figure 2 antibiotics-12-00476-f002:**
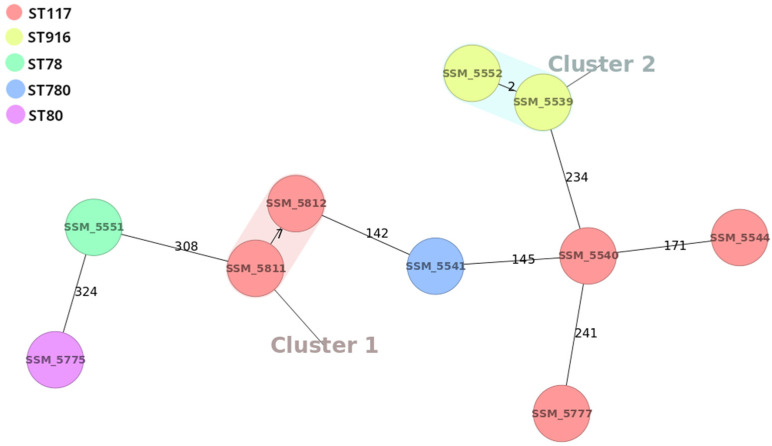
Minimum spanning tree generated by Ridom SeqSphere+ showing the genetic relatedness among *E. faecium* isolates. Each circle represents an isolate, and the color of each circle corresponds to the ST. The numbers on the connecting lines indicate the core gene SNP difference between strains.

**Table 1 antibiotics-12-00476-t001:** Characteristics of *E. faecium* isolates.

SSM Strain	Date	Source	Ward	AMP	GEN	STR	ERY	CIP	IMI	LNZ	TGC	VAN mg/L	TEC mg/L	Van Genes	ST
5538	4 February 2013	B. aspirate	Medicine	R	R	R	R	R	R	S	S	1	≤0.5	*-*	117
5550	6 March 2013	Urine	Medicine	R	S	R	R	R	R	S	S	≤0.5	≤0.5	*-*	117
5563	4 April 2013	Pus	Surgery	R	R	R	R	R	R	S	S	≤0.5	≤0.5	*-*	78
5556	7 April 2013	F. fluid	Surgery	R	R	R	R	R	R	S	S	≤0.5	≤0.5	*-*	78
5542	17April 2013	Sputum	Surgery	S	R	R	R	R	S	S	S	1	≤0.5	*-*	78
5540 *	22 April 2013	CVC	Surgery	R	R	R	R	R	R	S	S	≤0.5	≤0.5	*-*	117
5548	1 June 2013	Wound	ICU	R	R	R	R	R	R	S	S	1	≤0.5	*-*	78
5549	3 June 2013	Bile	Surgery	R	R	R	R	R	R	S	S	1	≤0.5	*-*	78
5551 *	6 June 2013	A. fluid	Surgery	R	R	R	R	R	R	S	S	1	≤0.5	*-*	78
5562	8 June 2013	Urine	Surgery	R	R	R	R	R	R	S	S	≤0.5	1	*-*	117
5555	1 July 2013	Blood	Medicine	R	R	R	R	R	R	S	S	1	≤0.5	*-*	78
5646	12 July 2013	Urine	Surgery	R	R	R	R	R	R	S	S	1	1	*-*	117
5560	19 July 2013	B. aspirate	Medicine	S	S	S	R	R	S	S	S	≤0.5	≤0.5	*-*	54
5539 *	4 August 2013	Bile	Surgery	R	S	R	R	R	R	S	S	≤0.5	≤0.5	*-*	916
5587	10 August 2013	D. fluid	Surgery	R	R	R	R	R	R	S	S	≤0.5	≤0.5	*-*	78
5588	10 August 2013	B. aspirate	Medicine	R	R	R	R	R	R	S	S	≤0.5	≤0.5	*-*	916
5583	28 September 2013	Biopsy	Medicine	R	R	R	R	R	R	S	S	≤0.5	≤0.5	*-*	117
5552 *	6 October 2013	Blood	Surgery	R	R	S	R	R	R	S	S	≤0.5	≤0.5	*-*	916
5591	2 November 2013	Bile	Surgery	R	R	R	R	R	R	S	S	≤0.5	≤0.5	*-*	78
5544 *	4 November 2013	B. aspirate	Surgery	R	R	R	R	R	R	S	S	≤0.25	≤0.25	*-*	117
5592	4 November 2013	Pus	Surgery	R	R	R	R	R	R	S	S	≤0.5	≤0.5	*-*	78
5541 *	4 December 2013	F. fluid	Surgery	R	R	R	R	R	R	S	S	1	≤0.12	*-*	780
5574	9 December 2013	Wound	Surgery	R	R	R	R	R	R	S	S	≤0.5	≤0.5	*-*	78
5652	28 December 2013	Blood	Medicine	R	R	R	R	R	R	S	S	≤0.5	≤0.5	*-*	780
5662	21 February 2014	A. fluid	Surgery	R	R	R	R	R	R	S	S	≤0.5	≤0.5	*-*	117
5670	26 February 2014	B. aspirate	ICU	R	R	R	R	R	R	S	S	≤0.5	≤0.5	*-*	117
5673	20 March 014	Blood	ICU	R	S	R	R	R	R	S	S	1	2	*-*	117
5671	27 March 2014	Blood	ICU	R	R	R	R	R	R	S	S	≤0.5	≤0.5	*-*	117
5653	2 April 2014	Blood	ICU	R	R	R	R	R	R	S	S	≤0.5	≤0.5	*-*	117
5655	1 May 2014	Bile	Surgery	R	R	R	R	R	R	S	S	1	≤0.5	*-*	78
5654	1 May 2014	Urine	Medicine	R	R	R	R	R	R	S	S	≤0.5	1	*-*	117
5658	2 May 2014	Urine	Medicine	R	R	R	R	R	R	S	S	≤0.5	2	*-*	117
5656	2 May 2014	Pus	ICU	S	S	S	R	R	S	S	S	≤0.5	≤0.5	*-*	74
5775 *	20 January 2017	D. fluid	Surgery	R	S	R	R	R	R	S	S	≤0.5	1	*-*	80
5777 *	25 January 2017	A. fluid	ICU	R	R	R	R	R	R	S	S	≤0.5	≤0.5	*vanB*	117
5779	9 February 2017	Bile	Surgery	R	S	R	R	R	R	S	S	≤0.5	2	*-*	117
5780	9 February 2017	Bile	Surgery	R	S	R	R	R	R	S	S	≤0.5	≤0.5	*-*	80
5811 *	2 May 2018	Wound	ICU	R	R	R	R	R	R	S	S	8	≤0.5	*vanB*	117
5812 *	3 May 2018	CVC	ICU	R	R	R	R	R	R	S	S	≥32	≤0.5	*vanB*	117

* These isolates were subjected to whole genome sequencing. CVC = central venous catheter; A. fluid: abdominal fluid; D. fluid: drainage fluid; F. fluid: fistula fluid; B. aspirate: broncho aspirate. The medicine ward included pathology and pneumology. AMP: ampicillin; GEN: high-level gentamicin; STR: high-level streptomycin; ERY: erythromycin; CIP: ciprofloxacin; IMI: imipenem; LNZ: linezolid; TGC: tigecycline; VAN: vancomycin; TEC: teicoplanin.

**Table 2 antibiotics-12-00476-t002:** Features and levels of vancomycin resistance mediated by *vanB*_2_-VREfm. Glycopeptides MICs are given before (A, B) and following (C) exposure to 100mg/L of vancomycin.

SSM Strain	Date	Source	Ward	A Vitek2		B MicroScan		C Vitek2	
				VAN mg/L	TEC mg/L	VAN mg/L	TEC mg/L	VAN mg/L	TEC mg/L
5777	25/01/2017	A. fluid	ICU	≤1	≤0.5	8	≤2	≥32	≤0.5
5811 *	02/05/2018	Wound	ICU	8	≤0.5	>8	≤2	≥32	≤0.5
5812 *	03/05/2018	CVC	ICU	≥32	≤0.5	>8	≤2	≥32	≤0.5

* Same patient.

## Data Availability

The data presented in this study are available in the article and in Appendix A.

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
