# Peer review of "Occult Vancomycin-Resistant Enterococcus faecium ST117 Displaying a Highly Mutated vanB2 Operon"

_antibiotics, 2023, doi:10.3390/antibiotics12030476_

Round 1

Reviewer 1 Report

In attached file

Author Response

Dear Reviewer 1,

Thanks for your valuable comments, we answered to your request as follow and also directly on PDF file renamed “Response to Reviewer 1 antibiotics-2201553-review.pdf” (Please see the attachment)

Page 1 Numerous abbreviations used in the abstract make it difficult to understand the text. Please avoid them

Response : It has been corrected

Page 3 Table 1 please expand the abbreviations for drugs

Response: We have added in  Table 1 legend

Page 5 

what does it mean:  "invasive or non-invasive infections"

Response: We meant that isolates were collected from blood or other samples. However we decided to delete the phrase since the specimen's origin is already described in Table 1.

What were the criteria for selecting the strains for the WGS? Were 3 isolates from your study and the rest from another source? If so, this should be indicated.

Response: We selected the isolates according to Hospital Associated Sequence Types, we corrected the phrase as follows "Ten genomes of E. faecium isolates were selected for WGS according to hospital associated ST. Five ST117, including the vanB-VREfm, two ST916 and one isolate from ST80, ST78 and ST780 were sequenced.”

Please explain the abbreviations used   

Response: It has been corrected

Page 6

Were these genes detected in all your isolates? It is not clear.

Response:  antibiotic resistance genes were detected just in sequenced isolates.  

We corrected as follows in the text (in red the additions): “Moreover, among the 10 sequenced E. faecium other resistance genes were detected in silico by ResFinder.

Abbreviations in parentheses should be expanded. (gyrA and parC substitution)

Response: Done Ser83-Tyr, Ser80-Ile

It would be necessary to indicate which strain possessed which set of genes

Response:  for major clarity we made an additional Supplementary Table S1b which include the characteristics of the 10 sequenced Enterococcus faecium isolates from University Hospital of Sassari, Italy in particular the resistant genes

please expand the abbreviations 

Response: Done

Page 8

please insert references for EUCAST

Response : We added reference [27] “The European Committee on Antimicrobial Susceptibility Testing. Breakpoint tables for interpretation of MICs and zone diameters. Version 12.0, 2022. http://www.eucast.org."

Susceptibility to what drugs was tested in this study? Please complete this information

Response : we integrated in MM. After identification, antimicrobial susceptibility testing was conducted by Vitek2 using the AST cards P592 and P658 (bioMerieux, Marcy-l'Étoile, France), which are specifically for gram positives Staphylococcus spp., Enterococcus spp. and Streptococcus agalactiae. For Enterococcus spp among the antibiotics tested we report results for ampicillin (AMP); high-level gentamicin (HI-GEN), high-level streptomycin (STR); ciprofloxacin (CIP); imipenem (IMI); erythromycin (ERY); linezolid (LNZ); teicoplanin (TEC) and vancomycin (VAN); tigecycline (TGC) following the European Committee on Antimicrobial Susceptibility Testing (EUCAST) guidelines [27].”

Please insert references for EUCAST

Response: we added in the References list

  1. The European Committee on Antimicrobial Susceptibility Testing. Breakpoint tables for interpretation of MICs and zone diameters. Version 12.0, 2022. http://www.eucast.org.

the abbreviation was explained in the introduction

Response: ok

This sentence is incomprehensible. Please edit them

Response: we re-edited as “ In this work, we highlighted a highly mutated vanB2-VREf ST117 isolate, occult by Vitek2 with the ability to increase its vancomycin MIC after exposure to vancomycin. This isolate was found to have similarities to both vanA-VVEfm and stealthy vanB2-E. faecium. 

Reviewer 2 Report

The author presented a good manuscript with quality results which needs a minor revision. The comments and suggestions are given below.

There are two abbreviations in Abstracts (VREfm and vanB2-VREfm), which need explanation. Also, there is an additional abbreviation for vancomycin MIC in the Abstract (VAN MIC), which appears excessive; MIC is enough in this context.

In the Introduction, the authors should comment on what a “teicoplanin” is before giving information on “the vanB operon is not induced by teicoplanin”.

The hospital where the E.faecium isolates were collected is referred to differently in the manuscript: “University hospital of Sassari” and “University hospital, AOU Sassari.”

In section 2.1, the authors should comment on “P592 and P658 cards”. What are the specifics of these cards?

In Figure 1, the authors should remove photo A with fingers and a bright spot and give only photos B-C after size correction and with a scale bar.

In section 2.1, the authors should specify “different methods” in the phrase “Glycopeptides MICs using the different methods are resumed in Table 2.” 

In section 2.2, the text in brackets “(Single Locus Variant of ST117 by E.burst)” needs to be explained, and maybe it will be better without brackets.

In section 2.3, there is a contradiction between numbers: “ten selected genomes were sequenced,” and only three biosamples were deposited. This contradiction needs to be explained. The IDs of two biosamples are given improperly.

In section 2.3, the authors give a list of amino acid substitutions. What is the possible impact of these mutations? Are they in variable or conservative parts of protein? Are they in some functional domains? Were they described before?

In section 2.3, the authors wrote that the other ten resistance genes were detected in silico by ResFinder. However, it is unclear: in what isolates among ten sequenced; how does it correspond to phenotypic resistance? Are they intrinsic for E.faecium or acquired?

In section 4.2.1, the reference needs to be added to the phrase “MLST was carried out on all isolates with the standard primers included in the E. faecium MLST scheme”.

In section 4.2.1, the brackets for the text “(DNA clean and concentratorTM-5-USA)” should be removed, and the text should be integrated into the phrase.

In section 4.2.2, the authors should describe how they prepared the WGS library.

In section 4.2.2, the authors mentioned Table 3, which is absent in the manuscript.

Author Response

Dear Reviewer 2,  

Thanks for your valuable comments and for your decision, we are grateful. Regarding your requests and suggestions we answered point by point as follows.

Point 1. There are two abbreviations in Abstracts (VREfm and vanB2-VREfm), which need explanation. Also, there is an additional abbreviation for vancomycin MIC in the Abstract (VAN MIC), which appears excessive; MIC is enough in this context.

Response 1: We have corrected by explaining Vancomycin-Resistant Enterococcus faecium (VREfm) and substituted vanB2-VREfm with vanB2 carrying VREfm. We have also omitted the VAN abbreviation as suggested.

Point 2. In the Introduction, the authors should comment on what a “teicoplanin” is before giving information on “the vanB operon is not induced by teicoplanin”.

Response 2: The introduction has been updated to include what is teicoplanin as follows ( in bold): “Unlike vanA, which confers inducible resistance to high levels of both glycopeptides vancomycin and teicoplanin, the vanB operon is not induced by teicoplanin and encodes for a variable level of inducible vancomycin resistance,...”

Point 3. The hospital where the E. faecium isolates were collected is referred to differently in the manuscript: “University hospital of Sassari” and “University hospital, AOU Sassari.”

Response 3: Yes, it might be confusing. We have mentioned throughout the manuscript University hospital of Sassari.

Point 4. In section 2.1, the authors should comment on “P592 and P658 cards”. What are the specifics of these cards? 

Response 4: AST-P592 and AST-P658 are both antibiogram cards for gram positives to be used in VITEK® 2 System for antimicrobial susceptibility testing of Staphylococcus spp., Enterococcus spp. and S. agalactiae. We specified in section 2.1    “………Vitek2 using the Gram-Positive Susceptibility Test Card P592 and P658 cards”. 

In section 4.1 MM we also have re-written as follows “After identification, antimicrobial susceptibility testing was conducted by Vitek2 using the AST cards P592 and P658 (bioMerieux, Marcy-l'Étoile, France), which are specifically for gram positives Staphylococcus spp., Enterococcus spp. and Streptococcus agalactiae. For Enterococcus spp among the antibiotics tested we report results for ampicillin (AMP); high-level gentamicin (HI-GEN), high-level streptomycin (STR); ciprofloxacin (CIP); imipenem (IMI); erythromycin (ERY); linezolid (LNZ); teicoplanin (TEC) and vancomycin (VAN); tigecycline (TGC) following the European Committee on Antimicrobial Susceptibility Testing (EUCAST) guidelines.

Point 5. In Figure 1, the authors should remove photo A with fingers and a bright spot and give only photos B-C after size correction and with a scale bar.

Response 5: We decided to delete Figure 1 in agreement with the suggestion of Reviewer 2 and incorporate in the Results section text 2.1 as follows: As a result of disk diffusion for vancomycin susceptibility in the SSM5777 isolate, a fuzzy inhibition zone edge was observed in the SSM5777 isolate. Despite the zone diameter of 12 mm, colonies were found within the inhibition zone, so the isolate was reported as resistant according to the notes of EUCAST Clinical Breakpoint.

Point 6. In section 2.1, the authors should specify “different methods” in the phrase “Glycopeptides MICs using the different methods are resumed in Table 2.” 

Response 6: We specified as follows: “Glycopeptides MICs were determined by both Vitek2 system and Microscan before exposure to vancomycin, and by Vitek2 following exposure to vancomycin, as resumed in Table 2.”

Point 7. In section 2.2, the text in brackets “(Single Locus Variant of ST117 by E.burst)” needs to be explained, and maybe it will be better without brackets. 

Response 7:  the sentence has been rephrased as follows: 

The ST117 (9,1,1,1,1,1,1,) was first detected in 2013, while the ST80 (9,1,1,1,12,1,1,), which  differs from ST117 at a single locus (gyd) by eBURSTv3  analysis, appeared in 2017 (Table 1).

Point 8. In section 2.3, there is a contradiction between numbers: “ten selected genomes were sequenced,” and only three biosamples were deposited. This contradiction needs to be explained. The IDs of two biosamples are given improperly. 

Response 8: Actually all the Biosamples have been deposited, two with a released status. For the rest the release will be available upon publication. For major clarity the accession numbers of the sequences, type of data and status are summarized in new generated Supplementary Table S1a in Supplementary Tables excel file.

Point 9.  In section 2.3, the authors give a list of amino acid substitutions. What is the possible impact of these mutations? Are they in variable or conservative parts of protein? Are they in some functional domains? Were they described before? 

Response 9: We have described it in the Discussion section as follows:

Our occult vanB2-isolate showed three mutations in the VanB2 protein in common with the other 2 vanB2VREfm. Two of these mutations, V321M and M308L, were described to be involved in the decrease of vancomycin MIC in vanB2-VREfm isolates from Japan and Taiwan, respectively [15, 19 ]. The presence of a VanS mutation (D105N) located in the same periplasmic domain [20] in reverted japanese isolates [15], may be responsible for the higher vancomycin MIC in our VREfm compared to the occult VREfm, lacking this mutation. Moreover, the analysis of the vanB2 operon highlighted additional mutations in VanW and VanH proteins, including a A231G missense mutation in VanH which occurs in an amino acidic residue essential for its dehydrogenase activity.

Point 10. In section 2.3, the authors wrote that the other ten resistance genes were detected in silico by ResFinder. However, it is unclear: in what isolates among ten sequenced; how does it correspond to phenotypic resistance? Are they intrinsic for E.faecium or acquired?

Response 10: The list of antimicrobial resistant genes detected by ResFinder in sequenced isolates is in a new Supplementary Table S1b. Presence of acquired antimicrobial resistant genes and phenotypes are in alignment.  

Point 11. In section 4.2.1, the reference needs to be added to the phrase “MLST was carried out on all isolates with the standard primers included in the E. faecium MLST scheme”.

Response 11: The proper reference has been added to the list : 29. Homan WL, Tribe D, Poznanski S, Li M, Hogg G, Spalburg E, Van Embden JD, Willems RJ. Multilocus sequence typing scheme for Enterococcus faecium. J Clin Microbiol. 2002 Jun;40(6):1963-71. doi: 10.1128/JCM.40.6.1963-1971.2002

Point 12. In section 4.2.1, the brackets for the text “(DNA clean and concentratorTM-5-USA)” should be removed, and the text should be integrated into the phrase.

Response 12: It has been Done

Point 13. In section 4.2.2, the authors should describe how they prepared the WGS library.

Response 13: Preparation of the DNA libraries was performed with the Nextera XT DNA Sample Preparation Kit (Illumina Inc. San Diego, CA, U.S.A.) and sequenced using the HiScanSQ (Illumina Inc.) with 93bp x 2 paired-end reads.

Point 14. In section 4.2.2, the authors mentioned Table 3, which is absent in the manuscript.

Response 14: We are really sorry, there must be some mistakes during submission uploading …

However, since we have created a Supplementary Tables file we included the list of the primers used for re-sequencing the vanB operon in Supplementary Table S1c. 

Reviewer 3 Report

- The discussion section should be expanded.

- Image 1 can be deleted

Author Response

Dear Reviewer 3

Thanks for your decision, we are grateful.

Regarding your request, we answered as follows

Point 1 The discussion section should be expanded.

Response 1: We agreed that this aspect is worth it. We integrated in the text your request by expanding the Discussion section in particular the part regarding the mutations (see revised manuscript).

Point 2.  Image 1 can be deleted 

Response 2: Yes, we have deleted Figure 1 in accordance with your suggestion and incorporated in the Results section 2.1 text as follows: “As a result of disk diffusion for vancomycin susceptibility in the SSM5777 isolate, a fuzzy inhibition zone edge was observed in the SSM5777 isolate. Despite the zone diameter of 12 mm, colonies were found within the inhibition zone, so the isolate was reported as resistant according to the notes of EUCAST Clinical Breakpoint”.